# Process Control Monitor (PCM) for Simultaneous Determination of the Piezoelectric Coefficients *d*_31_ and *d*_33_ of AlN and AlScN Thin Films

**DOI:** 10.3390/mi13040581

**Published:** 2022-04-07

**Authors:** Hao Zhang, Yang Wang, Lihao Wang, Yichen Liu, Hao Chen, Zhenyu Wu

**Affiliations:** 1State Key Laboratory of Transducer Technology, Shanghai Institute of Microsystem and Information Technology, Chinese Academy of Sciences, Shanghai 200050, China; zhanghao@mail.sim.ac.cn (H.Z.); wangyang@mail.sim.ac.cn (Y.W.); lhwang@mail.sim.ac.cn (L.W.); yichen_liu@mail.sim.ac.cn (Y.L.); haochen@mail.sim.ac.cn (H.C.); 2School of Graduate Study, University of Chinese Academy of Sciences, Beijing 100049, China; 3Shanghai Industrial μTechnology Research Institute, Shanghai 201800, China; 4School of Microelectronics, Shanghai University, Shanghai 200444, China

**Keywords:** Sc_x_Al_1−x_N, piezoelectric film, laser interferometry, piezoelectric constants, synchrotron X-ray diffraction

## Abstract

Accurate and efficient measurements of the piezoelectric properties of AlN and AlScN films are very important for the design and simulation of micro-electro-mechanical system (MEMS) sensors and actuator devices. In this study, a process control monitor (PCM) structure compatible with the device manufacturing process is designed to achieve accurate determination of the piezoelectric coefficients of MEMS devices. Double-beam laser interferometry (DBLI) and laser Doppler vibrometry (LDV) measurements are applied and combined with finite element method (FEM) simulations, and values of the piezoelectric parameters *d*_33_ and *d*_31_ are simultaneously extracted. The accuracy of *d*_31_ is verified directly by using a cantilever structure, and the accuracy of *d*_33_ is verified by in situ synchrotron radiation X-ray diffraction; the comparisons confirm the viability of the results obtained by the novel combination of LDV, DBLI and FEM techniques in this study.

## 1. Introduction

Piezoelectric thin films are widely used in various microelectromechanical systems (MEMS), such as communication devices [1], applications in the automotive industry [2], medical devices [3], and other integrated sensors and actuators [4,5]. These devices and systems are rapidly prevailing in the market [6]. Aluminum nitride (AlN) thin films have gained significance as piezoelectric materials due to their complementary metal-oxide semiconductor (CMOS) compatibility [7], high-temperature, long-term stability, high voltage resistance and good electrical insulation properties [8,9]. Moreover, research has shown that a significant increase in the piezoelectric constants *d*_33_ and *d*_31_ can be achieved by doping transition metals into AlN, with scandium (Sc) being the foremost option [10]. The characteristic parameters of piezoelectric materials not only influence the performance of MEMS but also affect their electromechanical conversion efficiency. To precisely predict the electromechanical performance of MEMS devices, the use of finite element method (FEM) software for device design requires accurate implementation of the elastic, dielectric and piezoelectric properties [11]. Therefore, accurate and efficient measurement of the characteristic parameters of piezoelectric films is important. Research has shown that the performance of these sensors and actuators is mainly related to the transverse piezoelectric coefficient *d*_31_ and the longitudinal piezoelectric coefficient *d*_33_ and is less affected by other parameters of the piezoelectric coefficient matrix, such as *d*_11_, *d*_12_, *d*_13_, *d*_14_, *d*_15_, *d*_16_, *d*_21_, *d*_22_, *d*_23_, *d*_24_, *d*_25_, *d*_26_, *d*_32_, *d*_34_, *d*_35_, and *d*_36_ [1,2,3,4,5].

There are various measurement techniques for piezoelectric thin films, such as using the direct piezoelectric effect by measuring the resulting voltage upon application of mechanical stress or using the inverse piezoelectric effect by measuring voltage-induced mechanical expansion or compression.

The formulas of the inverse piezoelectric effect and the positive piezoelectric effect of the longitudinal piezoelectric coefficient *d*_33_ are:(1)$$d_{33}={\left(\frac{\partial S_{3}}{\partial E_{3}}\right)}_{T}$$
(2)$$d_{33}={\left(\frac{\partial D_{3}}{\partial T_{3}}\right)}_{E}$$

The formulas of the inverse piezoelectric effect and the positive piezoelectric effect of the transverse piezoelectric coefficient *d*_31_ are:(3)$$d_{31}={\left(\frac{\partial S_{1}}{\partial E_{3}}\right)}_{T}$$
(4)$$d_{31}={\left(\frac{\partial D_{3}}{\partial T_{1}}\right)}_{E}$$
where *S* is strain; *E* is electric field; *D* is electrical displacement; *T* is stress; 1 is *x*-axis direction; and 3 is *z*-axis direction.

Lefki and Dormans proposed the direct piezoelectric measurement method and studied the ideal situation suitable for substrate clamping and interactions between electrode and substrate sizes. They showed that *d*_33_ obtained by the direct measurement method is the effective value *d*_33,*f*_ and affected by the clamping of the substrate. The relationship between *d*_33_ and *d*_33,*f*_ is given by [12]:(5)$$d_{33,f}=d_{33}-2d_{31}\frac{s_{13,p}}{s_{11,p}+s_{12,p}}$$
where *d*_33_ and *d*_31_ are the actual longitudinal and transverse piezoelectric coefficients, and *s*_11,*p*_, *s*_12,*p*_, and *s*_13,*p*_ are the compliance coefficients of the piezoelectric film.

The method of using a MEMS structure to extract piezoelectric properties has been thoroughly studied. In general, only one piezoelectric parameter (*d*_31_ or *d*_33_) can be obtained by the method mentioned above. For example, *d*_31_ can be extracted by using cantilever bending, which requires a dedicated process for cantilever release [13]. Since the performance of a film bulk acoustic resonator (FBAR) device is mainly affected by *d*_33_, the performance of a surface acoustic wave (SAW) device is mainly affected by *d*_31_. After measurement of the electrical and acoustic characteristics of a film, the piezoelectric co-efficient can be calculated; therefore, it is possible to use SAW resonators to extract *d*_31_ or FBARs to extract *d*_33_ [8,9,10,11,12,13,14]. However, when using this approach, various input parameters—such as the dielectric constant or sound velocity—should be determined before performing the electroacoustic characterization of fabricated FBAR and SAW devices, which introduces further uncertainties in their equivalent electrical circuit model.

In addition, the longitudinal deformation of a film can be measured by laser interferometry to characterize its longitudinal piezoelectric coefficient *d*_33_. Since the displacement of a piezoelectric film is very small (from pm to nm), optical interferometry, such as laser Doppler vibrometry (LDV) or double-beam laser interferometry (DBLI), is used to characterize the piezoelectric constant of the film (related to *d*_33_) [15]. When a piezoelectric film was deposited on a silicon wafer and not carefully clamped, Kholkin et al. found a quadratic relationship between the displacement measurement and the length of the electrode. A small electrode and hard conductive adhesive were used to inhibit substrate bending and improve test accuracy [16,17].

In general, it is difficult to simultaneously extract the piezoelectric coefficients *d*_31_ and *d*_33_ using a single test structure. A MEMS sensor based on piezoelectric films mainly utilizes the positive piezoelectric effect of the film during operation and realizes the sensing function by applying out-of-plane stress (related to *d*_33_) and then detecting the corresponding polarized charge. The actuator mainly uses the inverse piezoelectric effect of the film to apply an electric field in the polarization direction of the film to generate mechanical deformation in the longitudinal direction (related to *d*_33_) or transverse direction (related to *d*_31_) to drive the actuator. The performance of piezoelectric thin-film-based MEMS devices is usually affected by *d*_31_ and *d*_33_. Therefore, to accurately predict device performance, it is necessary to input accurate values of *d*_31_ and *d*_33_ during simulation analysis.

Recently, Mayrhofer et al. reported a 2-port electrode design based on the finite element (FEM) method and used it to measure the piezoelectric constant of AlScN films [17,18]. This measurement proved that it was possible to simultaneously extract the piezoelectric coefficients *d*_33_ and *d*_31_. However, they adopted empirical values from other studies as input parameters for their FEM simulations, which resulted in a larger amount of computation. The accuracy of their results has not yet been verified by other methods.

In this paper, we report a fabrication process for PCM test structures that does not require process steps in addition to patterning the AlScN and Mo electrodes. Due to the simplicity of the manufacturing process, the PCMs are compatible with most piezoelectric MEMS device fabrication processes. One great advantage lies in the significance of accurately defining the material parameters of the device, since the growth conditions of a piezoelectric film have a very obvious effect on its performance, and films grown in different batches have large performance differences. The DBLI method is used to pretest the longitudinal piezoelectric coefficient *d*_33_. It provides an intermediate reference value for the piezoelectric parameter scan in the simulation, which reduces the calculation time, shortens the bivariate iteration cycle and improves the extraction efficiency of the piezoelectric coefficient. The following sections of this paper characterize the material parameters of the two piezoelectric films (AlN and Al_0.8_Sc_0.2_N) used for the tests in this study, discuss how the PCM test structure was fabricated using the MEMS process, and explain how FEM simulations were efficiently used to simultaneously characterize the longitudinal (*d*_31_) and transverse (*d*_33_) piezoelectric constants. We then describe how we observed the reversible elongation and contraction of lattice parameters in situ under applied electric fields using synchrotron X-ray diffraction. The effective longitudinal piezoelectric constant *d*_33_ was estimated from the relationship between the electric field and the field-induced strain, which verified the *d*_33_ result obtained by the LDV-FEM method. The results for *d*_31_ were verified with the coefficients characterized by the cantilever method.

## 2. Materials and Methods

### 2.1. Preparation

A straightforward, 4-mask microfabrication process was developed for the fabrication of the PCM. Figure 1 shows the process of preparing a PCM on a 4-inch wafer. The top electrode was patterned by ion beam etching (IBE), and the 1-μm AlScN piezoelectric layer was wet etched with dilute tetramethylammonium hydroxide (TMAH) [19]. Since Sc doping increased the chemical resistance of the aluminum nitride film, it was necessary to increase the etching time for AlScN compared to undoped AlN. The top and bottom electrodes were electrically isolated by a SiO_2_ dielectric layer, followed by the growth of a Ti/Au metal layer, which was then patterned with IBE to obtain the shape of the test electrode for the PCM.

### 2.2. Physical Analysis

A focused ion beam (FIB) lamella was cut from the investigated PCM in the normal direction of the wafer surface to investigate the microstructure of the AlN and AlScN films. Figure 2a,b show that both the AlN and AlScN films had nearly perfect textured grain orientation along the direction normal to the substrate. Moreover, both films had sub-nm roughness, as identified from the clear sharp contrast for the Mo/AlN interfaces.

As a general rule, the FWHM of polycrystalline AlN thin films should be less than 2° to be used to fabricate high-performance MEMS devices. The X-ray diffraction (XRD) rocking curves can be found in Figure 2c,d for the AlN and AlScN films used in this study, from which the FWHM was found to be 1.32° for AlN and 1.53° for AlScN, respectively. Compared with the FWHM values reported in other studies [20,21], the results for the FWHM of the AlN and AlScN films used in this study confirmed that the piezoelectric film had a high c-axis orientation and good crystal quality.

Crystalline texture and diffraction images of AlN and AlScN were observed by selected area electron diffraction (SAD). The diffraction patterns in the SAD corresponded solely to the swurtzite-type AlN lattice in the highly oriented polycrystalline thin films, thus excluding the presence of metallic Sc crystal clusters. The AlN and AlScN layers both had a point diffraction pattern with (0001) as the preferred orientation parallel to the normal to the substrate (Figure 2e,f). The XRD and SAD results were in good agreement.

## 3. Results and Discussion

### 3.1. Piezoelectric Analysis

In this work, two top electrode designs were used for the DBLI test and LDV test, as shown in Figure 3a and Figure 4a, respectively. The square two-port electrode shown in Figure 4a was designed as an inner and outer two-layer electrode to reduce the deformation of the substrate by applying an AC voltage with a phase of 180° to the inner and outer electrodes, respectively (See Appendix A for details of the DBLI test). Additionally, the square electrode was extended in the longitudinal direction to increase the total electrode area, which reduced the influence of substrate deformation while realizing a larger excitation area of the piezoelectric layer. However, a square two-port electrode design with a larger area resulted in measurements that were much larger than the actual *d*_33_ value in the DBLI test. According to Sivaramakrishnan et al., the size of the top electrode should be similar to the substrate thickness to obtain consistent piezoelectric parameters [22]. Therefore, the diameter of the single circular electrode was set to 500 μm, which was similar to the thickness of the underlying silicon substrate, as shown in Figure 3a.

First, the longitudinal piezoelectric coefficient *d*_33_ of the piezoelectric thin films was extracted by DBLI (aixACCT Systems GmbH, Aachen, Germany). The test schematic diagram can be found in Figure 3. Optionally, a reflective thin film, such as Ti/Au, could be sputtered on the polished surface of the bottom side of the Si substrate to improve the quality of the test signal reflection [22].

A PCM with a single circular electrode design on the same wafer was characterized. Notably, DBLI measurement technology uses double laser beams to eliminate any contribution of substrate bending to *d*_33,*f*_ [22]. The device used in the DBLI test is shown in Figure 3a. The DBLI test was applied to the sample, and the results are shown in Figure 3c,d [23]. The results for *d*_33,*f*_ were obtained by fitting the average slope of the test results for voltage and displacement, *d*_33,*f*,*AlN*_ = 3.18 pm/V and *d*_33,*f*,*AlScN*_ = 5.18 pm/V, and the test results showed good repeatability (<3%).

Since the accuracy of the test results of DBLI in Figure 3 is not fully verified by other test method, in this study, we only used the *d*_33,*f*_ results from DBLI to refine the value ranges of the *d*_33_ and *d*_31_ simulation parameters in the FEM method, to more effectively obtain the accurate *d*_33_ and *d*_31_ results. For wurtzite type crystals with known *d*_33_ and *d*_31_ relationship [20]:−*d*_33_/2 < *d*_31_ < −*d*_33_/2 + 2.2 (pm/V)(6)

Substituting *s*_11,*p*_, *s*_12,*p*_, and *s*_13,*p*_ into Formula (5), we obtained *d*_33_ greater than *d*_33,*f*_ and less than twice the value of *d*_33,*f*_. Therefore, Formula (7) was obtained:d_33,f_ < d_33_ < 2d_33,f_(7)

Substituting *d*_33,*f*,*AlN*_ and *d*_33,*f*,*AlScN*_ from DBLI into Formula (7), we obtained 3.18 pm/V < *d*_33,*AlN*_ < 6.36 pm/V and 5.18 pm/V < *d*_33,*AlScN*_ < 10.28 pm/V. Substituting the above inequality into Formula (6), we obtained 5.18 pm/V < *d*_33,*AlScN*_ < 10.28 pm/V and −5.18 pm/V < *d*_31,*AlScN*_ < −2.98 pm/V. Therefore, the simulation parameter sweep range was set as shown in Table 1.

Then, the strain of the thin film material under an applied voltage was tested using LDV (Polytec MSA 500). A 180° phase shifted 65 kHz AC excitation voltage was amplified to 100 V by a high voltage amplifier (Aigtek ATA-2032) and applied to the inner and outer electrodes to excite the piezoelectric thin film and the out-of-plane component of the strain was recorded via LDV (See Appendix A for details of the LDV test). Because the test structure is a complete body structure, the resonance frequency of the test structure can be obtained by simulation, which is about GHz. The test frequency is far away from the resonance point, so the displacement response measured at this frequency reflects the real situation of film strain, and there is no possibility of resonance amplification. Piezoelectric constants were determined by subsequent FEM simulations, which were carried out with COMSOL software. All samples were attached to thick aluminum substrates with stiff insulating epoxy glue, and the aluminum substrates were fixed on the test bench to reduce the pulling effect of the surface deformation on the bottom surface of the substrate to a negligible level. 

Figure 4b shows the cross section of the PCM. The materials and dimensions of each layer can be found in Table 2. The 4-inch <100> silicon wafer consisted of a 450-μm silicon substrate and an AlN/AlScN layer with upper and lower Mo electrode interlayers. Table 3 shows the material parameters used in the FEM simulation. According to the calculation method in reference [24], *s*_11,*p*_, *s*_12,*p*_, and *s*_13,*p*_ were calculated from elastic constants. The calculation results are listed in Table 3.

Due to symmetry, the simulated structure consisted of two-dimensional boxes representing each layer, as shown in Figure 4c. The thickness of the piezoelectric film was composed of at least 15 quadrilateral elements to form a grid. The finite element calculation was performed under the condition that a voltage of U = 100 V was applied to the gold electrode on the upper surface of the AlScN layer. Figure 4c shows that the 2-port electrodes had opposite deflection displacements, so the substrate bending was reduced to a lower level. A small amount of the electrode displacement excited by the voltage was transferred to the aluminum substrate. Therefore, to improve the accuracy of determining the piezoelectric parameters, the simulation model of the inner and outer two-port electrodes included the aluminum substrate part. Finally, the complete scan result of the displacement along the vertical direction of the centerline on the surface of the piezoelectric film material was extracted into the data set for the measurement and evaluation of the piezoelectric constant.

Then, the electrode longitudinal deflection results were measured for the electrodes shown in Figure 4a. We estimated the piezoelectric coefficients of AlN and AlScN. Figure 5a,c compare the images of the deflection measurements and the finite element results under different sets of *d*_33_ and *d*_31_ parameter combinations.

The fitting between the test and the simulation data was performed by the least squares method. The fitting results for all parameter combinations were compared, and the simulation data for the parameter combination with the smallest calculation result were considered to be the best experimental result. To intuitively obtain the fitting results of each parameter combination, Figure 5b,d compare the least squares results of some parameter combinations. We found that *d*_33,*AlN*_ = 4.1 pm/V, *d*_31,*AlN*_ = −1.7 pm/V, *d*_33,*AlScN*_ = 9.9 pm/V and *d*_31,*AlScN*_ = −4.0 pm/V obtained the smallest fitting value. This means that this parameter combination was most consistent with the actual piezoelectric parameter values of the piezoelectric thin film. Therefore, the resulting piezoelectric coefficients for the AlN samples were *d*_33_ = 4.1 pm/V and *d*_31_ = −1.7 pm/V, agreeing excellent with those reported by Hernando et al. [15] and Mayrhofer et al. [18]. The resulting piezoelectric coefficients for the AlScN samples were *d*_33_ = 9.9 pm/V and *d*_31_ = −4.0 pm/V. These results agree well with earlier measurements of *d*_33_ of AlScN (*d*_33_ ~ 10 pm/V) reported by Akiyama et al. [11]. In addition, the values for *d*_33_ and *d*_31_ are in line with results from ab initio calculations that predict values for AlScN of about *d*_31_ ~ −5 pm/V and *d*_33_ ~ 10 pm/V [25].

Substituting the measured *d*_33_ and *d*_31_ results into Formula 5, we obtained *d*_33,*f*,*AlN*_ = 3.061 pm/V and *d*_33,*f*,*AlScN*_ = 7.220 pm/V, which were very similar to the *d*_33,*f*_ value measured by in situ XRD below.

### 3.2. Verification of the Results

To verify the accuracy of the piezoelectric parameters obtained by the abovementioned FEM method, we used synchrotron X-ray diffraction at the Shanghai Synchrotron Radiation Facility (SSRF) to observe the reversible elongation and contraction of the lattice parameters of the same batch of samples under an external electric field. The in-situ XRD experiments were carried out using synchrotron radiation with a 10.48 keV photon energy (*λ* = 0.12438 nm) performed on the BL02U beamline, with a beam diameter of 300 μm. Figure 6 shows the peaks for the AlN and AlScN films under different DC voltages (See Appendix A for details of the in-situ XRD test). As the negative voltage increased, the diffraction peak moved to a lower angle and vice versa.

The change in the lattice length under the action of the longitudinal electric field accurately represented the longitudinal piezoelectric coefficient *d*_33_ of the material. To obtain accurate calculation results, the film thickness shown in Figure 2a,b was used in the calculation.

The out-of-plane XRD pattern results are shown in Figure 6. According to the relationship between the electric field and the field-induced strain, the effective longitudinal piezoelectric constants *d*_33,*f*,*AlN*_ = 3.335 pm/V and *d*_33,*f*,*AlScN*_ = 7.560 pm/V were estimated; they were very close to the value of *d*_33,*f*_ calculated by our proposed method. At present, there is no team has characterized the *d*_33_ coefficient of AlN and AlScN films by synchrotron X-ray diffraction. However, according to the existing research, the accuracy of synchrotron X-ray diffraction test results has been well proved [26].

It’s worth noting that, the growth process of piezoelectric film inevitably introduces residual stress into the film, and according to the research of Berfield, T.A et al. [27], the magnitude of piezoelectric response will decrease with the increase of residual stress. In this paper, in-situ XRD is used to characterize the effective forward piezoelectric coefficient *d*_33_. Because the test results are obtained by applying different DC bias on the same sample for multiple measurements, the relationship between the obtained strain and the applied voltage is a relative value. Therefore, in-situ XRD can eliminate the influence of residual stress on piezoelectric test results.

SEM images of the cantilever beam structure used in the test and the material and thickness of each layer of the cantilever are shown in Figure 7a,b. The influence of the length and thickness of the cantilever on the measurement results has been given by Dekkers et al. [28]. We made 200 μm, 300 μm, 400 μm and 500 μm cantilevers and measured their displacement at resonance; the calculated *d*_31_ values are marked in Figure 7c (See Appendix A for details of the cantilevers test). We found that the law of the obtained *d*_31_ values was the same as that found by Dekkers et al., and the *d*_31_ result obtained by the 400 μm cantilever was closest to the true value. Formula 8 was used to obtain the effective transverse piezoelectric coefficients of the piezoelectric films: *d*_31,*AlN*_ = −1.68 pm/V and *d*_31,*AlScN*_ = −3.89 pm/V.
(8)$$\delta =\frac{3d_{31}s_{s}s_{p}t_{s}\left(t_{s}+t_{p}\right)L^{2}V}{s^{2}{}_{s}t^{4}{}_{p}+4s_{s}s_{p}t_{s}t^{3}{}_{p}+6s_{s}s_{p}t^{2}{}_{s}t^{2}{}_{p}+4s_{s}s_{p}t^{3}{}_{s}t_{p}+s_{p}t^{4}{}_{s}}$$
where *δ* is the cantilever displacement, *s_s_* and *s*_*p*_ are the mechanical compliances of the substrate and the AlScN, *t_s_* and *t_p_* are the thicknesses of the substrate and the AlScN respectively, *L* is the length of the cantilever and V is the excitation voltage. The compliance is related to Young’s modulus *E* by *s* = 1/*E*. The values for Young’s moduli are listed in Table 3. The material and thickness of each layer used in the calculation of the cantilever are listed in Table 4. The thickness of the Mo electrodes had little effect on the results, and it is given here to guide the fabrication of devices.

Our results are very close to those obtained by Stoecke et al. (*d*_31,*AlN*_ ~ −1.5 pm/V) [29]. The test results can provide reference value for the test and research of AlN cantilever beam method.

Table 5 shows that the difference between the *d*_31_ result obtained by the cantilever method and the *d*_31_ results obtained by the LDV method was within 10%, which verified the accuracy of the LDV method.

## 4. Conclusions

In this paper, we proposed a new test method that combined DBLI and LDV-FEM to improve the extraction efficiency of transverse and longitudinal piezoelectric coefficients. The on-chip integrated process control monitor (PCM) test structure was fabricated with the flow sheet of the MEMS device without creating a discrete test structure. This quick approach used few MEMS process steps to simultaneously evaluate the piezoelectric coefficients of the AlN and AlScN thin films. The pure AlN and 20% scandium-doped AlScN films grown by reactive magnetron sputtering were tested; the longitudinal effective piezoelectric coefficients were *d*_33,*AlN*_ = 4.1 pm/V and *d*_33,*AlScN*_ = 9.9 pm/V, and the transverse effective piezoelectric coefficients were *d*_31,*AlN*_ = −1.7 pm/V and *d*_31,*AlScN*_ = −4.0 pm/V. In situ XRD measurements of the film were conducted using the Shanghai synchrotron radiation source to verify the accuracy of the *d*_33_ results, and the cantilever method was used to verify the accuracy of the *d*_31_ results. As shown in Table 5, the errors of the various test methods were within 10%, which proved the accuracy of our new method. This work provided a reliable and efficient method for the characterization of piezoelectric films and strong support for the design and simulation of Sc_x_Al_1__−x_N-based piezoelectric MEMS devices with enhanced electromechanical properties.

## Figures and Tables

**Figure 1 micromachines-13-00581-f001:**
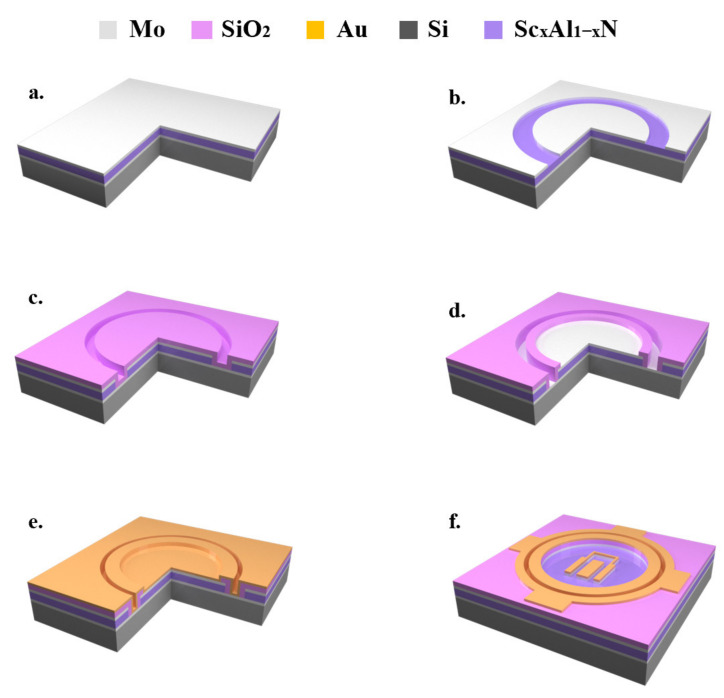
The process flow of the PCM structure. (**a**) a stack of Mo/AlN(AlScN)/Mo is deposited by magnetron sputtering; (**b**) a 0.2 µm Mo top layer is patterned and etched as a hard mask for the underlying AlN/AlScN layer etching; (**c**) a 1 µm layer of SiO_2_ is deposited by PECVD; (**d**) the top and bottom electrodes are exposed after SiO_2_ patterning; (**e**) a 0.5 µm layer of Ti/Au was deposited; (**f**) the top Au electrode was patterned for electrical connection.

**Figure 2 micromachines-13-00581-f002:**
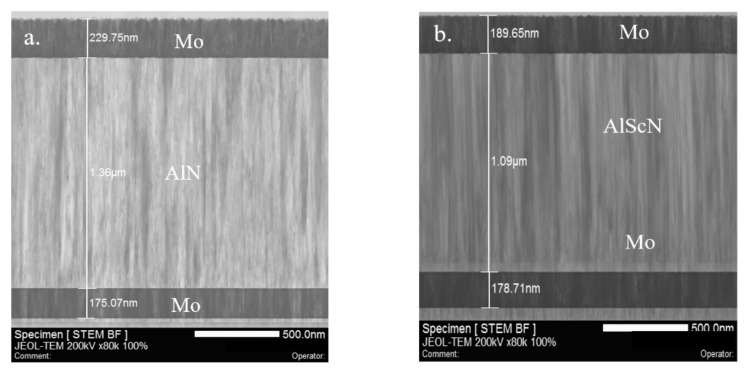
(**a**) Scanning transmission electron microscopy (STEM) image of the Mo-AlN-Mo structure; (**b**) STEM image of the Mo-AlScN-Mo structure; (**c**) Rocking curves of the AlN film; (**d**) rocking curves of the AlScN film; (**e**) selected area diffraction (SAD) pattern of AlN; (**f**) SAD pattern of AlScN.

**Figure 3 micromachines-13-00581-f003:**
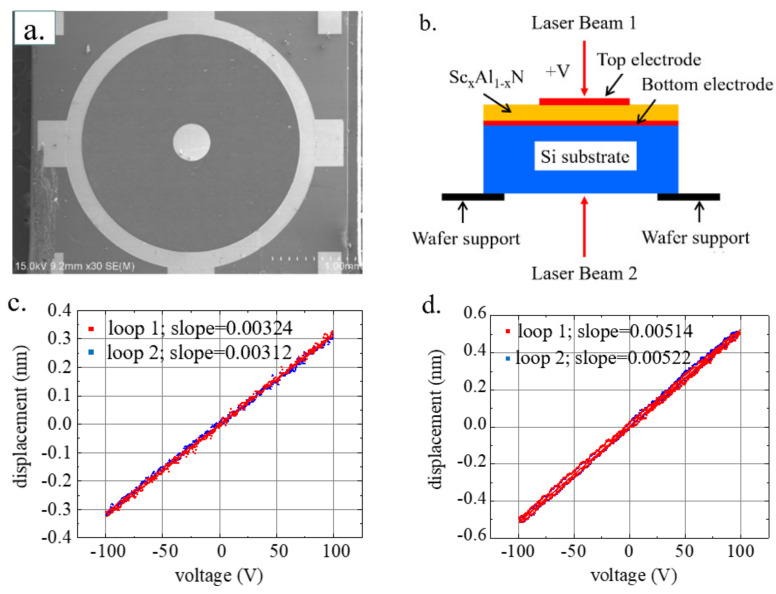
(**a**) The scanning electron microscope (SEM) image of the device used in DBLI test. (**b**) Schematic diagram of DBLI test system and test principle; (**c**) DBLI test results of AlN and (**d**) AlScN.

**Figure 4 micromachines-13-00581-f004:**
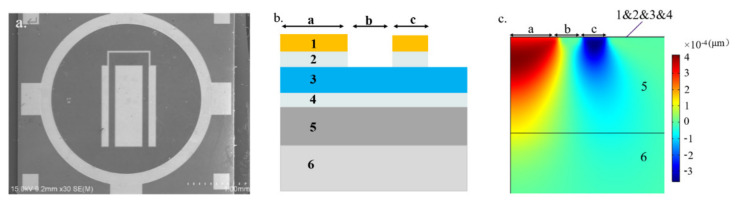
(**a**) SEM image of the device used in the LDV test; (**b**) 2D cross-sectional view of the 2-port electrode design used as input for the FEM simulations; (**c**) FEM results for the local displacement perpendicular to the wafer surface.

**Figure 5 micromachines-13-00581-f005:**
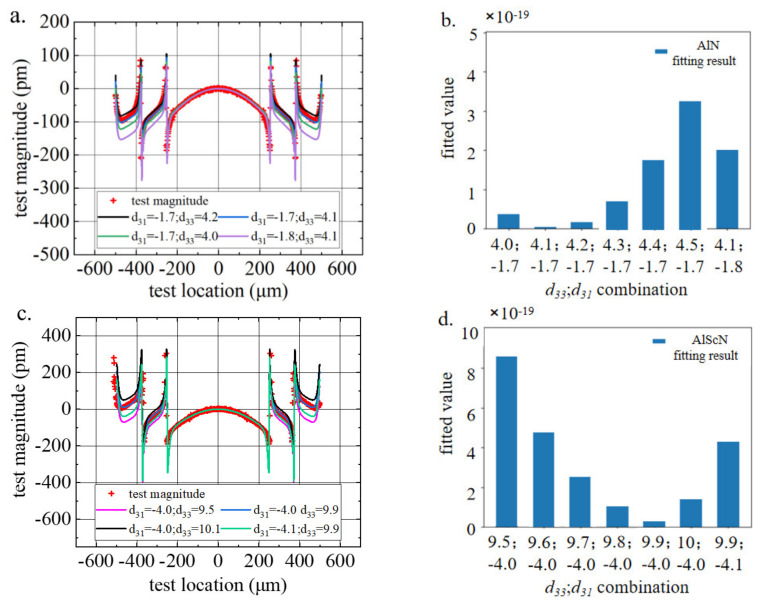
(**a**,**b**) AlN and (**c**,**d**) AlScN test and simulation data fitting results.

**Figure 6 micromachines-13-00581-f006:**
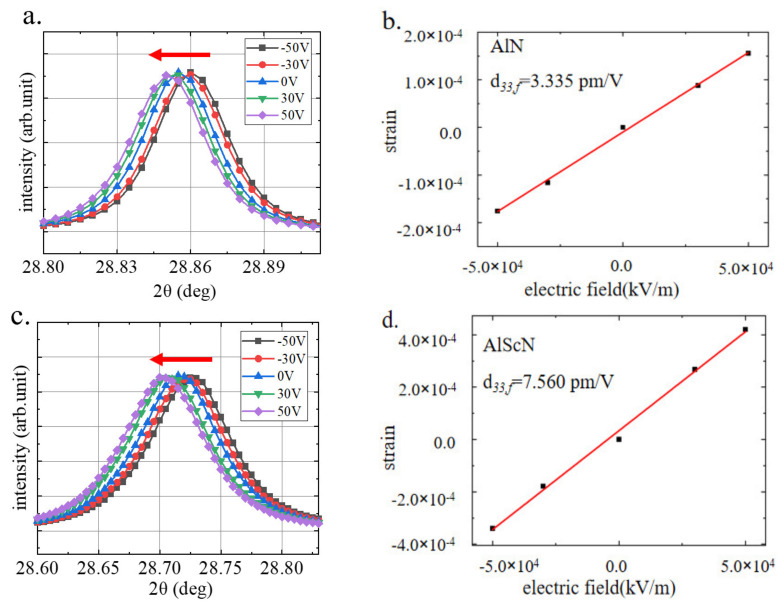
Out-of-plane XRD patterns of the (**a**) AlN and (**b**) AlScN thin films obtained for the (002) peak under varying DC voltages; The *d*_33,*f*_ linear fit results of (**c**) AlN and (**d**) AlScN thin films.

**Figure 7 micromachines-13-00581-f007:**
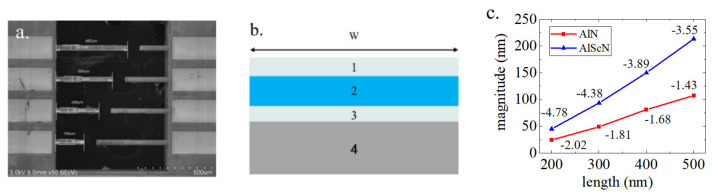
(**a**) SEM image of the cantilever beam; (**b**) the test and simulation results for AlN and AlScN cantilever beams with different lengths of AlN and AlScN; (**c**) the test results of the AlN and AlScN cantilevers.

**Table 1 micromachines-13-00581-t001:** The simulation parameter range settings.

Material	Range of *d*_31_ [pm/V]	Range of *d*_33_ [pm/V]	Step Length [pm/V]
AlN	−3.18 to −0.98	3.18 to 6.36	0.1
AlScN	−5.18 to −2.98	5.18 to 10.28	0.1

**Table 2 micromachines-13-00581-t002:** Input parameters of the simulation model for the FEM.

Layer	Material	Thickness [nm]	Gap	Width [μm]
1	Au	200 nm	a	250 μm
2	Mo	200 nm	b	125 μm
3	AlN/AlScN	1.36 μm/1.09 μm	c	125 μm
4	Mo	200 nm	/	/
5	Si	450 μm	/	/
6	Al	2 mm	/	/

**Table 3 micromachines-13-00581-t003:** The material parameters used in FEM simulation.

Material	Property	Value	Units
Anisotropic Si <100>	Young’s modulus	130	[GPa]
	Poisson’s ratio	0.28	/
AlN [24]	Young’s modulus	338	[GPa]
	*c*_11_ = *c*_22_	345	[GPa]
	*c* _33_	395	[GPa]
	*c* _12_	125	[GPa]
	*c* _13_	120	[GPa]
	*c*_44_ = *c*_55_	118	[GPa]
	*c*_66_ = (*c*_11_ − *c*_12_)/2	110	[GPa]
	*s*_11_ = *s*_22_	3.53	[10^−12^ m^2^/N]
	*s* _12_	−1.01	[10^−12^ m^2^/N]
	*s* _13_	−0.77	[10^−12^ m^2^/N]
	*s* _33_	3	[10^−12^ m^2^/N]
	*s*_44_ = *s*_55_	8.48	[10^−12^ m^2^/N]
	*s* _66_	9.09	[10^−12^ m^2^/N]
Sc_0.2_Al_0.8_N [18]	Young’s modulus	230	[GPa]
	*c*_11_ = *c*_22_	325	[GPa]
	*c* _33_	279	[GPa]
	*c* _12_	138	[GPa]
	*c* _13_	131	[GPa]
	*c*_44_ = *c*_55_	99	[GPa]
	*c*_66_ = (*c*_11_ − *c*_12_)/2	94	[GPa]
	*s*_11_ = *s*_22_	4.14	[10^−12^ m^2^/N]
	*s* _12_	−1.02	[10^−12^ m^2^/N]
	*s* _13_	−1.38	[10^−12^ m^2^/N]
	*s* _33_	4.88	[10^−12^ m^2^/N]
	*s*_44_ = *s*_55_	10.1	[10^−12^ m^2^/N]
	*s* _66_	10.6	[10^−12^ m^2^/N]

**Table 4 micromachines-13-00581-t004:** The material and thickness of each layer used in the calculation of the cantilever.

Layer	Material	Thickness	Width (W)
1	Mo	200 [nm]	35 [μm]
2	AlN (AlScN)	1.36 [μm] (1.09 [μm])	35 [μm]
3	Mo	200 [nm]	35 [μm]
4	Si	30 [μm]	35 [μm]

**Table 5 micromachines-13-00581-t005:** Comparison of different test methods.

	Cantilever Method [pm/V]	In-Situ XRD [pm/V]	LDV-FEM [pm/V]
AlN *d*_33,*f*_	/	3.335	3.061
AlN *d*_31_	−1.68	/	−1.7
AlScN *d*_33,*f*_	/	7.560	7.22
AlScN *d*_31_	−3.89	/	−4.0

## Data Availability

Data are available from the authors on request.

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
