# Peer review of "Process Control Monitor (PCM) for Simultaneous Determination of the Piezoelectric Coefficients d31 and d33 of AlN and AlScN Thin Films"

_micromachines, 2022, doi:10.3390/mi13040581_

Round 1
Reviewer 1 Report
The work is interesting, I recommend its acceptance after the response to my comments:
1- Please compare your results with others in the literature to show their additional value.
2 Please provide some details of the used X-ray beam energy in the synchrotron.
Author Response
Thank you very much for your valuable comments on the article. According to your comments, the modifications are as follows:
- Please compare your results with others in the literature to show their additional value.
Our response:
Thank you for you for your valuable advice.
The test results of LDV-FEM, in-situ XRD and cantilever beam are compared with those of other current research literature, and the reference value of our test results are explained. At present, there is no literature on AlN and AlScN testing with DBLI in the same field, so the comparison results of DBLI have not been given. In this paper, the test results of DBLI in our work only provide a reference value for the LDV-FEM test method to improve the simulation efficiency.
2. Please provide some details of the used X-ray beam energy in the synchrotron.
Our response:
Thank you for you for your valuable advice.
The X-ray energy used in the synchrotron radiation accelerator has been added to the supplementary materials and main body of the article.

Reviewer 2 Report
A process control monitor (PCM) structure is designed to achieve d31 and d33 simultaneously for MEMS devices. The method has some nolties. The manuscript can be accepted after minor revision.
1. d33,f is already obtained by DBLI test in Fig 3. (3.18 for AlN, 5.18 for AlScN). Why the d33,f is measured again in Fig.6? As said in the text above Fig.6, the d33,f values (3.061 for AlN, 7.220 for AlScN) are different?So the DBLI test is only a rough estimation?
2. How the residucal stress affect the measurment?
3. Figure 4 is not necessary. LDV is a well-known equipment. Since AC voltage is applied, what is the frequency of AC? How the frequency affect the results?
4. Enlgish writting needs improvement.
Author Response
Thank you very much for your valuable advices on the article. According to your comments, the modifications are as follows:
- d33,f is already obtained by DBLI test in Fig 3. (3.18 for AlN, 5.18 for AlScN). Why the d33,f is measured again in Fig.6? As said in the text above Fig.6, the d33,f values (3.061 for AlN, 7.220 for AlScN) are different?So the DBLI test is only a rough estimation?
Our response:
Thank you for you for your valuable advice.
Since the accuracy of the test results of DBLI in Figure 3 is not fully verified by other test data, only the test results of DBLI are used in this paper to provide reference values for the selection of simulation parameter range of LDV-FEM, so as to improve the efficiency of simulation analysis. In Figure 7, the d33,f obtained by the in-situ XRD measurement are recognized as more accurate results. Therefore, in the following, the accuracy of LDV-FEM measurement is verified by the results of in-situ XRD. And this description has been added to the article and marked in red.
- How the residucal stress affect the measurment?
Our response:
Thank you for you for your valuable advice.
The growth process of piezoelectric film inevitably introduces residual stress into the film, and according to the research of T. A. Berfield et al, the magnitude of piezoelectric response will decrease with the increase of residual stress. In this paper, in-situ XRD is used to characterize the effective forward piezoelectric coefficient d33. Because the test results are obtained by applying different DC bias on the same sample for multiple measurements, the relationship between the obtained strain and the applied voltage is a relative value. Therefore, in-situ XRD can eliminate the influence of residual stress on piezoelectric test results. In this paper, the influence of residual stress on the piezoelectric response of thin films has been explained according to the research of T. A. Berfield et al, and the test results are not affected by residual stress has been explained in the content of in-situ XRD test and marked in red.
- Figure 4 is not necessary. LDV is a well-known equipment. Since AC voltage is applied, what is the frequency of AC? How the frequency affect the results?
Our response:
Thank you for you for your valuable advice.
According to the suggestion, figure 4 has been deleted. The AC signal frequency applied in LDV test experiment is 65khz. A description has been added in the supplementary material and main body of the article. Because the test structure is a complete body structure, the resonance frequency of the test structure can be obtained by simulation, which is about GHz. The test frequency is far away from the resonance point, so the displacement response measured at this frequency reflects the real situation of film strain, and there is no possibility of resonance amplification. This part of the explanation has been added to the article and marked in red.
- Enlgish writting needs improvement.
Our response:
Thank you for you for your valuable advice.
English writing has been optimized. The reference content of each pronoun in the text is clarified to prevent unclear reference. The logical problem of the sentence in this paper is corrected. For example, change "this novel method" to "The novel combination of LDV, DBLI and FEM techniques in this study", Change "Calculation amount" to "Calculation time", and so on.
